# Sampling bias and model choice in continuous phylogeography: Getting lost on a random walk

Antanas Kalkauskas[1�¤], Umberto Perron[1☯], Yuxuan Sun[1], Nick Goldman[1], Guy Baele[2], Stephane Guindon[3], Nicola De Maio[1]*

**1** European Molecular Biology Laboratory, European Bioinformatics Institute, Hinxton, United Kingdom, **2** Department of Microbiology, Immunology and Transplantation, Rega Institute, KU Leuven, Leuven, Belgium, **3** Department of Computer Science, LIRMM, CNRS and Université de Montpellier, Montpellier, France

☯ These authors contributed equally to this work.
¤ Current address: Sensmetry, Vilnius, Lithuania
* demaio@ebi.ac.uk

**Data Availability Statement:** All scripts and data used for simulations and analyses can be found at https://github.com/NicolaDM/Phylogeography.

## Abstract

Phylogeographic inference allows reconstruction of past geographical spread of pathogens or living organisms by integrating genetic and geographic data. A popular model in continuous phylogeography—with location data provided in the form of latitude and longitude coordinates—describes spread as a Brownian motion (Brownian Motion Phylogeography, BMP) in continuous space and time, akin to similar models of continuous trait evolution. Here, we show that reconstructions using this model can be strongly affected by sampling biases, such as the lack of sampling from certain areas. As an attempt to reduce the effects of sampling bias on BMP, we consider the addition of sequence-free samples from under-sampled areas. While this approach alleviates the effects of sampling bias, in most scenarios this will not be a viable option due to the need for prior knowledge of an outbreak's spatial distribution. We therefore consider an alternative model, the spatial Λ-Fleming-Viot process (ΛFV), which has recently gained popularity in population genetics. Despite the ΛFV's robustness to sampling biases, we find that the different assumptions of the ΛFV and BMP models result in different applicabilities, with the ΛFV being more appropriate for scenarios of endemic spread, and BMP being more appropriate for recent outbreaks or colonizations.

## Author summary

Phylogeography studies past location and migration using information from current geographic locations of genetic sequences. For example, phylogeography can be used to reconstruct the history of geographical spread of an outbreak using the genetic sequences of the pathogen collected at different times and locations. Here, we investigate the effects of different model assumptions on phylogeographic inference. In particular, we examine the effects of the strategy used to collect samples. We show that sample collection biases can have a strong impact on the quality of phylogeographic reconstruction: geographically

**Funding:** AK and YS were supported by the Cambridge Mathematics Placements (CMP, https://www.maths.cam.ac.uk/opportunities/careers-for-mathematicians/summer-research-mathematics/summer-research-mathematics-cmp-and-research-cms). GB was supported by the Interne Fondsen KU Leuven / Internal Funds KU Leuven (https://www.kuleuven.be/english/research/support/if) under grant agreement C14/18/094, and from the Research Foundation – Flanders ('Fonds voor Wetenschappelijk Onderzoek – Vlaanderen', https://www.fwo.be/G0E1420N). SG was supported by the Agence Nationale pour la Recherche https://anr.fr/ through the grant GENOSPACE. AK, UP, YS, NG and NDM were supported by the European Molecular Biology Laboratory. The funders had no role in study design, data collection and analysis, decision to publish, or preparation of the manuscript.

**Competing interests:** The authors have declared that no competing interests exist.

biased sampling scheme can be very detrimental for popular continuous phylogeography models. We consider different ways to counter these effects, from utilising alternative phylogeographic models, to the inclusion of partially informative samples (known cases without genetic sequences). While these strategies do alleviate the effects of sampling biases, they also lead to considerable additional computational burden. We also investigate the intrinsic differences of different phylogeographic models, and their effects on reconstructed patterns in different scenarios.

## 1 Introduction

Genetic data can be very informative of migration histories and spatial patterns of living organisms, and of geographic spread of outbreaks, in particular when combined with information regarding present and past geographic ranges. Phylogeography combines genetic and geographic data to study geographical spread; in the context of geographic spread of outbreaks, which we will focus on in this manuscript, phylogeography often interprets observed genetic sequences as the result of sequence evolution along an evolutionary phylogenetic tree (see [1]), while modeling spatial spread as a separate evolutionary process along the same phylogeny (see e.g. [2–8]).

In recent years, Bayesian phylogeographic inference has gained remarkable popularity, in large part due to convenient implementations such as in the Bayesian phylogenetic inference software package BEAST [9, 10]. Bayesian phylogeography in BEAST allows users to investigate past geographical spread using genetic sequences possibly collected at different times. Genetic data is integrated with geographical and temporal sampling information, and all data is interpreted jointly in terms of evolution along a phylogenetic tree with heterochronous leaves [6, 7, 11–15]. BEAST uses Markov chain Monte Carlo (MCMC) to efficiently sample from the joint parameter space—which can also include parameters related to demographic reconstruction and phenotypic trait evolution—and in doing so, accurately accounts for uncertainty in phylogeny and model parameters, and possibly uncertainty in sampling time and location.

Bayesian phylogeographic approaches in BEAST can be divided into two categories depending on the type of geographical data: discrete space phylogeography and continuous space phylogeography. Discrete space phylogeography is typically used when samples are clustered based on their geographic location; this is appropriate when spread within a geographical unit is more or less free, while spread between units is hindered by geographical or political barriers (such as bodies of water, mountain chains, national borders, etc). In this case, the geographical data for a collected sample consists of a discrete geographical unit (e.g. a country). Oftentimes, the use of discrete phylogeography is one of necessity, e.g. when only the country of origin of the collected samples is known. Evolution of this location over time (e.g. spread between countries) along the phylogeny is usually modeled using a continuous-time Markov chain (see [6, 11]), similarly to popular phylogenetic models of sequence evolution (see [1]).

On the other hand, when the longitudinal and latitudinal coordinates of the samples are known, and when spread is assumed to happen more or less in a geographically homogeneous way over some area (such as on one island, or within one continent), continuous space phylogeography is often employed as an alternative to binning samples into discrete locations, which can be biasing [16]. In continuous phylogeography one typically models geographical spread along the branches of the tree as a Brownian motion process, which can be thought of as consisting of many small movements in random directions over many short time intervals

(see [7, 14]). The results of continuous phylogeography can subsequently be used to determine factors causing non-homogeneous spatial spread through space [17, 18].

A problem of discrete space phylogeography is that sampling bias (samples not being collected across locations proportionally to their prevalence) can strongly affect statistical inference [13]. Unbiased sampling can be very hard to achieve, as it requires knowledge of the full geographic range of an outbreak, access to the whole of this range, and extensive sampling and sequencing efforts. An alternative is to use models that are not affected, or less affected, by sampling biases, such as the structured coalescent and its approximations (see [12, 13, 15], although note that these can be adversely affected by unsampled or unknown demes [19–21]). The structured coalescent model, however, is far more computationally demanding than classical discrete space phylogeography and can differ from it on several aspects other than sampling assumptions. For example, the structured coalescent assumes that the migratory process and the distribution of cases across locations are at equilibrium, but these assumptions are rarely met in practice and do not match outbreaks that recently expanded into new areas.

Here, we investigate the effect of sampling biases in continuous space phylogeography. We show that sampling only certain areas of an outbreak can result in strongly inaccurate inference of dispersal history and the related model parameters. A possible alternative to the Brownian motion phylogeography ("BMP") model used in continuous space phylogeography is the spatial Λ-Fleming-Viot process ("ΛFV") recently introduced in population genetics (see [16, 22–26]). The ΛFV addresses, among other things, the undesirable equilibrium properties of classical models of geographic spread [27]. The ΛFV represents an alternative to the BMP, which would be expected to be mostly robust to sampling bias. We here show that the BMP and the ΛFV are non-interchangeable models, which are suitable for very different evolutionary scenarios. We also investigate the use of "sequence-free" samples (samples without genetic information) as a means to correct or help diagnose the effects of sampling biases on BMP.

## 2 Materials and methods

We assume that $N$ samples $s_1, \ldots, s_N$ have been collected, and each sample $s_i$ is associated with a genetic sequence $S_i$, a collection time $t_i$, and a location of collection $l_i \in \mathbb{R}^2$. Location $l_i$ is made up of longitude $l_i^{(1)}$ and latitude $l_i^{(2)}$, and represents the location of the sample at the time $t_i$ of collection. Sequence $S_i$ represents the genome (or part of the genome) of the sample, and usually provides most of the phylogenetic information. We assume that the phylogenetic tree $\tau$ is a time-stamped phylogeny, where the dates of the tips are known (corresponding to the collection times $t_i$) and can differ from each other; branch lengths are represented in units of time.

Our main focus is to infer the history of geographical spread, represented in particular here by the reconstruction of the location of the root node of $\tau$, and to infer the parameters of the migration process itself. We use two models to simulate and infer the migration process in continuous space: Brownian motion phylogeography (BMP) and the spatial Λ-Fleming-Viot process (ΛFV). Below we describe both models in detail.

### 2.1 Brownian motion phylogeography (BMP)

BMP assumes that changes in location happen along branches of $\tau$ according to a time-homogeneous Brownian (Wiener) diffusion process [28, 29]. Given any branch $b$ of length $t$ in $\tau$, and assuming that we know the location $l = (l^{(1)}, l^{(2)})$ of the parent node of this branch, then, under the BMP, the distribution of potential locations of the child node of $b$ is centered on $l$ and is multivariate normally distributed with variance $t\mathbf{P}^{-1}$, where $\mathbf{P}$ is the precision matrix of the BMP. In other words, conditional on the parent node of $b$ being in position $l$, the location

of the descendant node of the branch has distribution $\mathcal{N}_2(l, t\mathbf{P}^{-1})$. We assume that the precision matrix $\mathbf{P}$ is the same for all branches, and has three free parameters: two marginal precisions, and the correlation coefficient between dimensions. These parameters describe respectively how fast spatial movement happens in each dimension and how correlated the movements are in the two dimensions. For simplicity, we assume no changes in diffusion rates across branches, although we recognize that variation in diffusion rates is important in many real-life scenarios [7].

Under the BMP, the posterior probability of a set of parameters $\tau$ (the phylogeny), $\mathbf{\Theta}$ (the parameters describing sequence evolution along $\tau$), and $\mathbf{P}$ (the precision matrix of the BMP) conditional on the data $t_1, \ldots, t_N, S_1, \ldots, S_N, l_1, \ldots, l_N$ is:

$$P(\tau, \mathbf{\Theta}, \mathbf{P} | t_1, \ldots, t_N, S_1, \ldots, S_N, l_1, \ldots, l_N)$$

$$= \frac{P(\mathbf{\Theta})P(\mathbf{P})P(\tau, t_1, \ldots, t_N)P(S_1, \ldots, S_N | \tau, t_1, \ldots, t_N, \mathbf{\Theta})P(l_1, \ldots, l_N | \tau, t_1, \ldots, t_N, \mathbf{P})}{P(t_1, \ldots, t_N, S_1, \ldots, S_N, l_1, \ldots, l_N)} \quad (1)$$

This means that, given $\tau$ and $\mathbf{P}$, the migratory history (and therefore the observed locations) is independent of genetic data and evolution. Similarly, given $\tau$ and $\mathbf{\Theta}$, sequence evolution (and therefore observed sequences) is independent of geographic data and migratory process. It is usually not feasible to calculate the probability of the data (known as the marginal likelihood, or the normalizing constant), $P(t_1, \ldots, t_N, S_1, \ldots, S_N, l_1, \ldots, l_N)$, which appears in the denominator above. Instead, BEAST employs MCMC to obtain samples from the posterior density of model parameters without the need to calculate this probability. The terms in the numerator are:

- the prior $P(\mathbf{P})$ on the precision matrix $\mathbf{P}$ (usually a Wishart distribution [7]), and the prior $P(\mathbf{\Theta})$ on the substitution model and parameters $\mathbf{\Theta}$. For $P(\mathbf{\Theta})$, many choices are possible, depending on prior information available regarding the mutational process, and the models considered [30].

- the tree prior $P(\tau, t_1, \ldots, t_N)$ which represents the prior probability of observing a given tree and sampling times. Possible priors can be based on birth-death models [31] or coalescent models [32] (note however that for coalescent priors a different notation from Eq 1 is required, conditioning all probabilities on the sampling times $t_1, \ldots, t_N$).

- the classical phylogenetic likelihood $P(S_1, \ldots, S_N | \tau, t_1, \ldots, t_N, \mathbf{\Theta})$ that depends on a specific substitution model and parameters $\mathbf{\Theta}$ and that can be calculated using Felsenstein's pruning algorithm [33].

- the geographic likelihood $P(l_1, \ldots, l_N | \tau, t_1, \ldots, t_N, \mathbf{P})$ is the probability of the geographic locations given the precision matrix $\mathbf{P}$ and tree. This can be efficiently calculated by integrating out the location of internal tree nodes, similarly to Felsenstein's pruning algorithm but for a continuous trait [14, 34]. Some approaches opt for Gibbs sampling the ancestral node locations, for example in the work of [7]; in such cases, the notation of Eq 1 needs to be slightly modified.

There are a number of features that distinguish the BMP from the $\Lambda$FV presented in the next section, which are important to keep in mind. In Fig 1A we give a graphical representation of the BMP, and we here provide a short summary of the features of the model:

- BMP assumes that the prior probability of the tree $\tau$ is not affected by the migration process $\mathbf{P}$. Note however that the posterior probability of the tree might instead be very much affected by the geographic migration model and parameters.

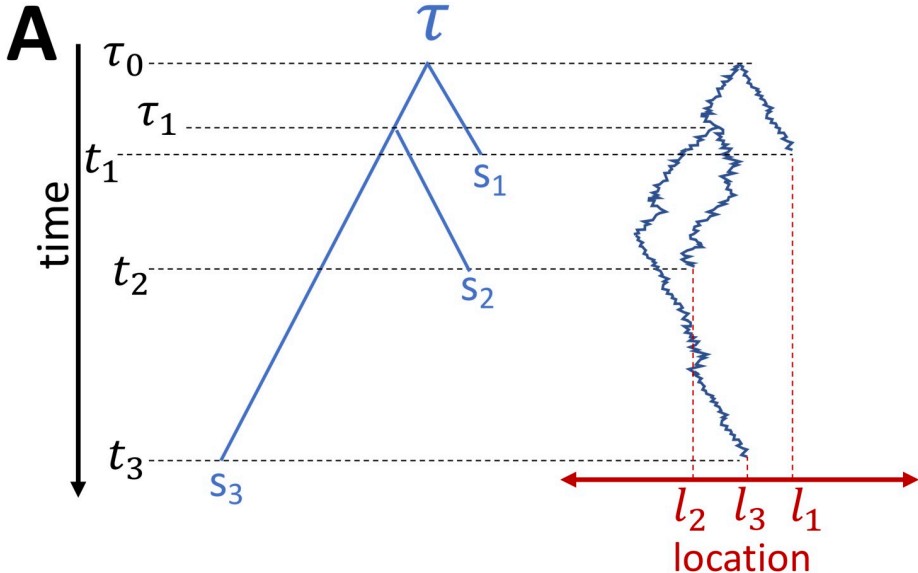

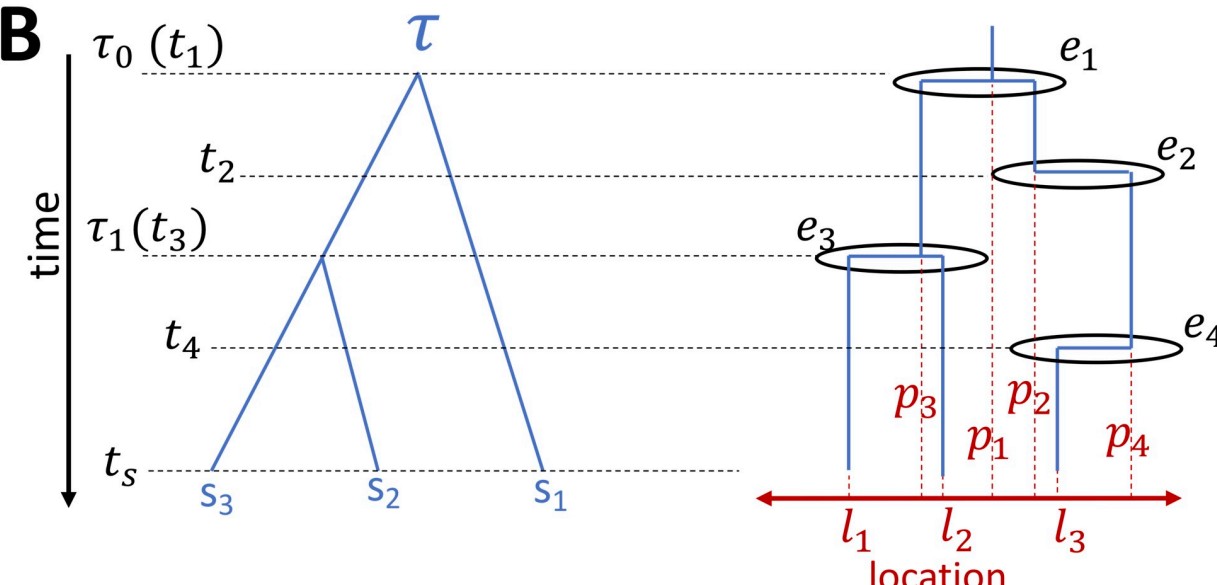

**Fig 1. Graphical example of BMP and ΛFV models.** Here we compare a graphical representation of the BMP model (**A**) against a graphical representation of the ΛFV (**B**). In both cases, time (black) is on the Y axis, with the forward direction pointing down, and $\tau$ (blue) is the phylogeny relating three samples $s_1$, $s_2$ and $s_3$. In **A**, samples are collected at different times ($t_1$, $t_2$ and $t_3$ respectively), while in **B** all samples are collected at the same time $t_s$, reflecting the ultrametric tree constraint of the ΛFV in its current simulation and inference software implementations. The time of the most recent common ancestor of $s_2$ and $s_3$ is $\tau_1$, while the time of the root of $\tau$ is $\tau_0$. On the right side of each plot we represent, for simplicity, a 1-dimensional space (red) on the X axis, instead of the 2-dimensional space we actually use for simulation and inference. $l_1$, $l_2$ and $l_3$ are the locations where the three samples $s_1$, $s_2$ and $s_3$ are collected. (The X axis positions of $s_1$, $s_2$ and $s_3$ within $\tau$ are however not meaningful, as in typical phylogenetic graphical representations.) The right-side diagram of **A** graphically mimics how the location of lineages changes along the phylogeny and along time as it evolves according to a Brownian motion. The right-side diagram of **B** similarly shows graphically how the location of lineages can change according to a ΛFV model; events like $e_1$ and $e_3$ can cause lineages to coalesce (backward in time), that is, to find a common ancestor, with the location of the parent lineage (respectively $p_1$ and $p_3$) being different from the locations of the descendant lineages. Other events, like $e_2$ and $e_4$, might result in only changes of location for a lineage, which moves (backward in time) to the location of the parent (respectively $p_2$ and $p_4$). Other events, not represented here, might not result in any change of ancestry or location of the ancestral lineages of the considered samples.

- BMP normally does not assume boundaries on possible geographic locations, so sample and ancestral node coordinates can be anywhere in the considered space (including in bodies of water, for example). Prior ancestral root locations can also be specified, see [35], and a normal prior distribution over root location is typically assumed, see [7]).

- BMP does not assume that the density of the overall population of cases over space and over time is uniform or at equilibrium, and does not aim to describe, at least explicitly, the migratory and reproductive dynamics of the whole population, but only of the ancestral lineages of the considered samples. It assumes instead that there is no interaction among cases (for example, limited resources or susceptible individuals within one area), so that different lineages evolve and spread independently of each other no matter how close they are in geographic space.

- in BMP, sampling locations are considered a result of pathogen spread, and not an arbitrary choice of the investigator. As such, sampling locations, even in the absence of genetic sequences, can be very informative about the process of geographic spread, as it is assumed that sampling locations are representative of the geographic range of the pathogen. This also means that absence of samples from certain areas will be interpreted by the model as evidence of absence of cases from such areas. In practice, if the sampling process is dependent on geography, for example when cases from some areas are more likely to be sampled than cases from other areas, then the inference under BMP can be affected, as we show below. This should not necessarily be considered a negative aspect of the model: if there is no sampling bias, then considering sampling locations as informative of the process of geographic spread can increase the inference power of the model.

- Currently, no backward-in-time descriptions of the BMP exist; such a description of a dual process of the BMP could be useful for performing BMP inference while avoiding assumptions about (and therefore biases from) the sampling process.

## 2.2 Spatial Λ-Fleming-Viot process (ΛFV)

The ΛFV can be used to model migration and evolution of individuals within a population distributed across an area. The geographical area $A$ under consideration is usually a torus (as in the simulator *discsim* [26]), or a rectangle (as in the phylogeographic inference software PhyREX [16]). Migration is only allowed from and into $A$, potentially representing, for example, the case of an island or a continental mass. Individuals of the population are assumed to be spread over $A$ with uniform density $\rho$. Migration and reproduction of individuals are modeled through reproduction-extinction events (from now on, just "events") which happen at rate $\lambda$ over time. Each event $e_i$ happening at time $t_i$ is centered at a location $c_i$ taken at random uniformly from $A$. Individuals in the population are affected by the event according to their distance from $c_i$. For example, in *discsim* all individuals within a radius $r$ around $c_i$ are affected, while in PhyREX individuals are affected with a probability that decreases with their distance from $c_i$ (specifically, according to a normal kernel with variance $\theta^2$). Individuals affected by $e_i$ then die with probability $\mu$, and new individuals are born around $c_i$. In the case of disc events (as in *discsim*), new individuals are born uniformly within the event disc with density $\rho\mu$. In the case of normal kernel events, as in PhyREX, new individuals are similarly placed so to leave the population distribution uniform. Lastly, one (or more in case of recombination [26, 36]) parents for all the individuals born at $e_i$ are chosen, again with a probability that decreases as a function of the distance from $c_i$ (again, either uniformly on a disc as in *discsim* or with a normal kernel as in PhyREX, for example).

While the ΛFV is very different from the BMP, some aspects of the two models can be compared. For example, for narrow event kernels (i.e. small $\theta$), the position of one lineage along one dimension after time $t$ is approximately normally distributed with variance $t\sigma^2$, where $\sigma^2 = 4\pi\theta^4\lambda\mu/|A|$ and $|A|$ is the area of $A$ [16], and at the limit of very small $\theta$ and very large $\lambda$, the movements of individuals approach a Brownian motion with diffusion rate $\sigma^2$. Similarly, in the case of disc events of radius $r$, the mean per-dimension diffusion rate approaches $\sigma^2 = \frac{\pi r^4 \lambda \mu}{2|A|}$ (see S1 Text).

Despite the fact that for small and frequent events individuals might move almost in a Brownian motion, there are still significant differences between the ΛFV and the BMP. The posterior probability of ΛFV model parameters (which we collectively represent as $\Lambda$), of a certain history $E$ of events $E = \{e_1, \ldots, e_{|E|}\}$, of tree $\tau$, and of substitution model parameters $\Theta$ is:

$$P(\tau, \Theta, \Lambda, E | t_1, \ldots, t_N, S_1, \ldots, S_N, l_1, \ldots, l_N)$$
$$= \frac{P(\Theta)P(\Lambda)P(S_1, \ldots, S_N | \tau, t_1, \ldots, t_N, \Theta)P(\tau, E | \Lambda, t_1, \ldots, t_N, l_1, \ldots, l_N)}{P(S_1, \ldots, S_N | t_1, \ldots, t_N, l_1, \ldots, l_N)}. \tag{2}$$

Similarly to the BMP, samples from the joint posterior density of model parameters can be obtained using MCMC, as is done by PhyREX. The terms in the numerator are:

- the prior $P(\Lambda)$ on the ΛFV model parameters, and the prior $P(\Theta)$ on the substitution model parameters $\Theta$.

- the classical phylogenetic likelihood $P(S_1, \ldots, S_N | \tau, t_1, \ldots, t_N, \Theta)$, as in the BMP.

- the likelihood of the history of events, and the ancestry and ancestral locations of the samples $P(\tau, E | \Lambda, t_1, \ldots, t_N, l_1, \ldots, l_N)$, which can be computed following [16].

In Fig 1B we give a graphical representation of the ΛFV. From Eq 2 and the description of the model, a number of differences with the BMP can be noted, of which we again provide a summary here:

- in the ΛFV, the probability of a tree $\tau$ can be affected by the spatial dynamics of the model.

- the ΛFV is defined over a finite space, and is hence more appropriate at describing migration within a limited area (such as an island or continent).

- the ΛFV assumes that the spatial density of the population is homogeneous and at equilibrium. This means that the model describes the case where resources are homogeneously spread across the environment, and the pathogen or species is endemic within an area (this excludes recent colonizations or recent outbreaks where the pathogen has not yet spread across the whole area).

- calculating the likelihood of the ΛFV, at least in implementations proposed so far [16, 37], requires the explicit parameterization of individual events. This means that inference under this model is typically going to be more computationally demanding than inference under the BMP, except for scenarios with very few events.

- the ΛFV always conditions on sampling times and sampling locations (see Eq 2). This is because, while the population is assumed homogeneously distributed through time and space, the sampling process is assumed to be arbitrary and not reflective or related to the density of the population or the migratory history. As such, the ΛFV should not be affected by any sampling bias.

- the ΛFV model has a backward-in-time dual process [16, 26]. This process describes the distribution of past events given data collected later on (see term $P(\tau, E|l_1, \ldots, l_N, t_1, \ldots, t_N)$ above), thereby naturally accommodating possible spatial sampling biases.

## 3 Results

### 3.1 Sampling biases in BMP

To investigate the effect of sampling bias on BMP, we simulated evolution and migration under the same BMP model used for inference, and tested different sampling scenarios. We simulated a Yule phylogenetic tree with birth rate 1.0, and stopped the simulations when 1000 tips were generated. Genetic sequences were assumed 10kb long, and we simulated their evolution using an HKY model ($\kappa$ = 3 and uniform nucleotide frequencies) and a substitution rate of 0.01 per unit time, ensuring reasonable levels of genetic diversity to allow reliable phylogenetic inference. Trees and sequences were simulated using DendroPy [38]. Using a custom python script, we simulated migration along the tree under the BMP model with two independent dimensions each with diffusion rate equal to 1 unit of square distance per time unit, and we always placed the root in (0.0, 0.0).

   Of the 1000 tips in the total tree (representing all the cases in the considered outbreak), we sampled 50 tips (representing the samples collected and sequenced) under four different strategies to simulate different types of sampling bias:

- in the first scenario ("Random Sampling"), samples were collected independently of their location, and as such no bias is expected and there is no model mis-specification;

- in the second scenario ("Central Sampling"), the closest samples to the source of the outbreak (0.0, 0.0) were collected;

- in the third scenario ("Diagonal Sampling"), the samples closest to the $x = y$ diagonal were collected;

- in the fourth and last scenario ("One-Sided Sampling"), the samples with the highest X coordinate (the most eastern samples) were collected.

   The sampling biases we simulate here can be considered extreme, and may not represent real sampling scenarios, but they showcase the effects that different types of sampling biases can have on phylogeographic inference.

   We used BEAST v1.10.4 to perform inference under the classical BMP model [7], assuming the default priors in BEAUti. During inference we did not restrict the two diffusion processes in the two dimensions to be independent or of equal rate, and inferred the correlation in the two diffusion processes and their rates. During both inference and simulations we assumed a constant rate migration process (see [7]). We ran the MCMC for $10^7$ steps and sampled the posterior every 1000 steps, which was sufficient to reach convergence (ESS much higher than 200, checked using Tracer [39]). We ran 100 simulated replicates, and we analysed each replicate four times according to the four sampling scenarios above. Under these four sampling scenarios, we find at least moderate correlation between samples' geographic distances and genetic distances: averages over 100 simulations of 0.192 for random sampling, 0.042 for central sampling, 0.176 for diagonal sampling, and 0.260 for one-sided sampling. This suggests that, in all scenarios, at least a moderate amount of signal to estimate geographic spread is present in the generated data.

   We found that the sampling strategy affects root location inference using BMP (Fig 2A and 2D and S1 Fig). In the absence of sampling bias, inference appears accurate (unbiased and

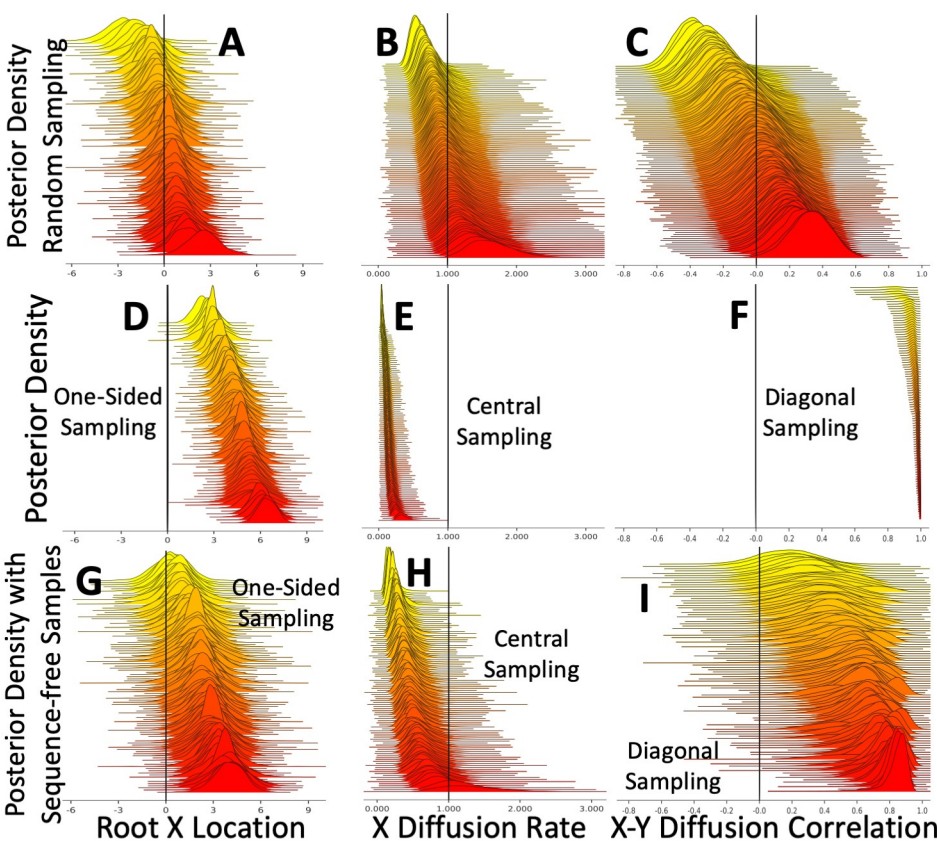

**Fig 2. bf Effects of sampling bias on BMP inference.** Here, a BMP model was used both for simulation and inference. Plots show inferred posterior distributions for the root X coordinate (plots **A,D,G**), the diffusion rate along the X dimension (plots **B,E,H**), and the correlation between the diffusion in the two dimensions (plots **C,F,I**). In each plot, the 100 distributions represent 100 independent replicates, which are vertically sorted based on the posterior median. Vertical black lines show the true simulated values. Plots **A-C** are from simulations with non-biased sampling. Plots **D-F** are respectively with "One-sided" sampling bias, "Central" sampling bias, and "Diagonal" sampling bias. Plots **G-I** are like **D-F** but with the addition of 50 sequence-free samples (see Section 3.2) collected independently of their geographic location. When plotting root locations, since in many cases the MRCA (root) of the collected samples is not the root of the whole simulated phylogeny (which is always located at (0.0, 0.0)), in each replicate all posterior locations are translated (in mathematical sense) so that the true MRCA location is always at (0.0, 0.0).

calibrated, Fig 2A). With central or diagonal sampling bias, the uncertainty and error of root location is further reduced (S1(C)–S1(F) Fig), but this probably reflects the fact that samples were collected close to the true origin of the simulated outbreak. When collecting samples at one extreme end of the outbreak, instead, we found that root location inference is strongly biased, with posterior distributions usually not containing the true simulated origin locations (Fig 2D).

The effects of sampling bias on the inference of BMP migration parameters are even more noticeable (Fig 2B, 2C, 2E and 2F and S2 Fig). While inference of diffusion rate with no sampling bias is correct and calibrated (Fig 2B), in every biased sampling scenario it is underestimated. In particular, with central and diagonal sampling the posterior distributions usually do not contain the true value (Fig 2E and S2 Fig). The reason for this is probably that the small sampled range (compared to the actual range of the outbreak) is interpreted as evidence of a small outbreak range, and therefore as low diffusion rate (absence of samples in an area interpreted as absence of cases). In the case of diagonal sampling, BMP also infers a strong

correlation between the migration processes in the two dimensions (the true value of 0 covariance is never covered by the posterior distributions, Fig 2F).

To test the effects of tree uncertainty and sequence data, we also ran inference under the scenario that the simulated tree is perfectly known, representing the case in which sufficient genetic information is available so that there is negligible uncertainty in tree inference. We provided no input alignment and specified no phylogenetic likelihood or substitution model, but instead fixed the tree to the simulated one and removed all transition kernels in BEAST that affect the tree. In this case, our analyses required substantially fewer MCMC steps ($10^4$), with parameters sampled every 10 steps. We found virtually identical results as those presented in Fig 2 (S3, S4 and S5 Figs).

We also performed an additional set of simulations under the One-Sided Sampling scenario, but this time with different levels of proportions of biased samples, so to investigate the effects of the sampling bias intensity. We simulated a phylogeny of 10,000 cases under BMP and we sampled 100 of them under 5 different intensities of one-sided sampling bias. At 100% intensity, only samples with the highest X coordinate are collected. At 75% intensity, we collected 75 samples with the highest X coordinate and 25 at random. Similarly for 50% and 25% intensities. At 0% intensities, 100 samples are collected uniformly at random. We find that at 75% bias intensity already a large part of the inference bias alreay disappears (S6 Fig). Further lowering the intensity of the bias reduces inference bias, but with diminishing returns. It is surprising to see that, at intermediate sampling bias intensities, diffusion rate on the X dimension is not underestimated (as at 100% bias intensity) but is rather slightly overestimated.

## 3.2 Compensating the effects of sampling biases using sequence-free samples

The biases shown above originate from the fact that the BMP assumes that samples are collected independently of location, and so the absence of samples from an area is evidence—for the BMP—of absence of cases in that area. Here, we explore the possibility of compensating for the effects of sampling bias in BMP by adding "sequence-free" samples to the analyses. This is representative of the case, for example, that we know that an outbreak has spread into a location, and we know the time and place of some of the cases in that location, but we cannot collect or sequence samples from those cases; so, some of the samples will be "proper", that is, will encompass genetic sequences, while the other "sequence-free" samples will have sampling location and time, but no genetic sequence (see also [40, 41]).

To recreate this scenario, we used the 100 Yule trees simulated before. As before, from each simulation, we considered 50 tips sampled according to the four sampling scenarios, representing "proper" samples with genetic sequence. Then, we selected another 50 sequence-free tips randomly (and independently of location) from the remaining 950 tips. These other 50 sequences were added to the BEAST analyses (for a total of 100 samples per replicate) without sequence data (or, more precisely, with uninformative sequences made only of gap characters "-") but with correct sampling location and date.

Adding these extra sequence-free samples greatly reduces the effects of sampling biases, but does not eliminate them (Fig 2G–2I, S7 and S8 Figs). To further investigate how many extra sequence-free samples would be needed to compensate for sampling biases, we simulated a One-Sided Sampling scenario with a reduced number of cases (100, so to make it computationally feasible to add most of them as sequence-free samples) and with 20 samples. We then perform inference either with no added sequence-free samples, or with a number of extra sequence-free samples between 20 and 80. Adding 80 extra samples means considering all cases simulated during inference, but only considering the sequences of the 20 biased samples.

We find that, in this scenario, adding 20 extra sequence-free samples removes most of the biases, and adding more than 40 extra sequence-free samples in this scenario brings no clear advantage (S9 Fig) while it considerably increases computational demands. This suggests that, while a considerable number of sequence-free samples is needed to compensate for sampling bias, it generally seems not necessary to add as many samples as to make the geographical sampling unbiased.

### 3.3 Can the ΛFV correct the sampling bias in BMP?

As mentioned before, the ΛFV has a number of differences from the BMP. One of these differences is that the model does not assume that the sampled locations are representative of the range of the outbreak, but instead the model assumes uniform density of cases over a considered, limited space. For this reason, the ΛFV should not be affected by sampling bias (see "Models" Section). We performed inference using the software PhyREX within the package PhyML v3.3.20190909 [16] downloaded on 4th of January 2020 from https://github.com/stephaneguindon/phyml.git. PhyREX implements the ΛFV model on a rectangular space. We fixed the tree to the simulated true one to greatly reduce the parameter space to be explored and to consider the case in which tree uncertainty is negligible (for example due to abundant genetic data). We used PhyREX to infer the diffusion rate ($\sigma^2$) of the migration process (see S1 Text) and the migration histories, together with the other parameters of the ΛFV model. We ran each PhyREX replicate analysis for 1 week or a maximum of $2 \times 10^8$ MCMC steps, sampling every 2000 steps. This seemed generally sufficient to reach convergence in all scenarios and most replicates; however, we note that achieving convergence under the ΛFV was considerably harder than under the BMP, probably due to the larger number of free parameters and the considerable uncertainty in their values. In 57 out of 600 replicate runs of PhyREX, at least one of the considered parameters had an effective sample size (ESS) below 100. We show below results both including and excluding these non-converged replicates.

First, we considered the same exact 1000-tips simulated Yule trees as described above, with BMP migration, four different sampling bias scenarios and 50 collected tips. The ΛFV model used for inference might now be very different from the BMP model used for simulations, and so model mis-specification could have a considerable impact. One important difference is that the BMP has no spatial boundaries by default, while the ΛFV is defined over a finite space. In PhyREX, we define the geographical space (where the migration process takes place) to be a square with dimension double the maximum coordinate of any simulated outbreak case, and centered in (0.0, 0.0), so that all simulated samples are contained within the considered square. We find results from PhyREX to be very different from those in BEAST. Credible intervals of the root location are now much broader, and always contain the truth (Fig 3A, S10 and S11 Figs). On the other hand, the diffusion rate is highly overestimated, up to hundreds of times, and the corresponding posterior distributions usually do not contain the truth (Fig 3B, S12 and S13 Figs). The large uncertainty in the root location is probably caused by the fact that the ΛFV model uses less information than the BMP (by not assuming that sampling locations are representative of prevalence) and is less affected by sampling bias; however, the high inferred diffusion rates suggest that model mis-specification also plays a strong role in these analyses. We found that setting a prior on the radius parameter so as to mimic BMP (i.e., migration events preferentially taking place over short distances) can reduce this bias.

To further investigate the differences between the BMP and ΛFV models, we simulated trees and migration under the ΛFV model implemented in *discsim* [26]. The ΛFV models in *discsim* and PhyREX differ in some aspects. One difference is that *discsim* assumes that death and recolonization events happen uniformly over discs, while PhyREX uses normal

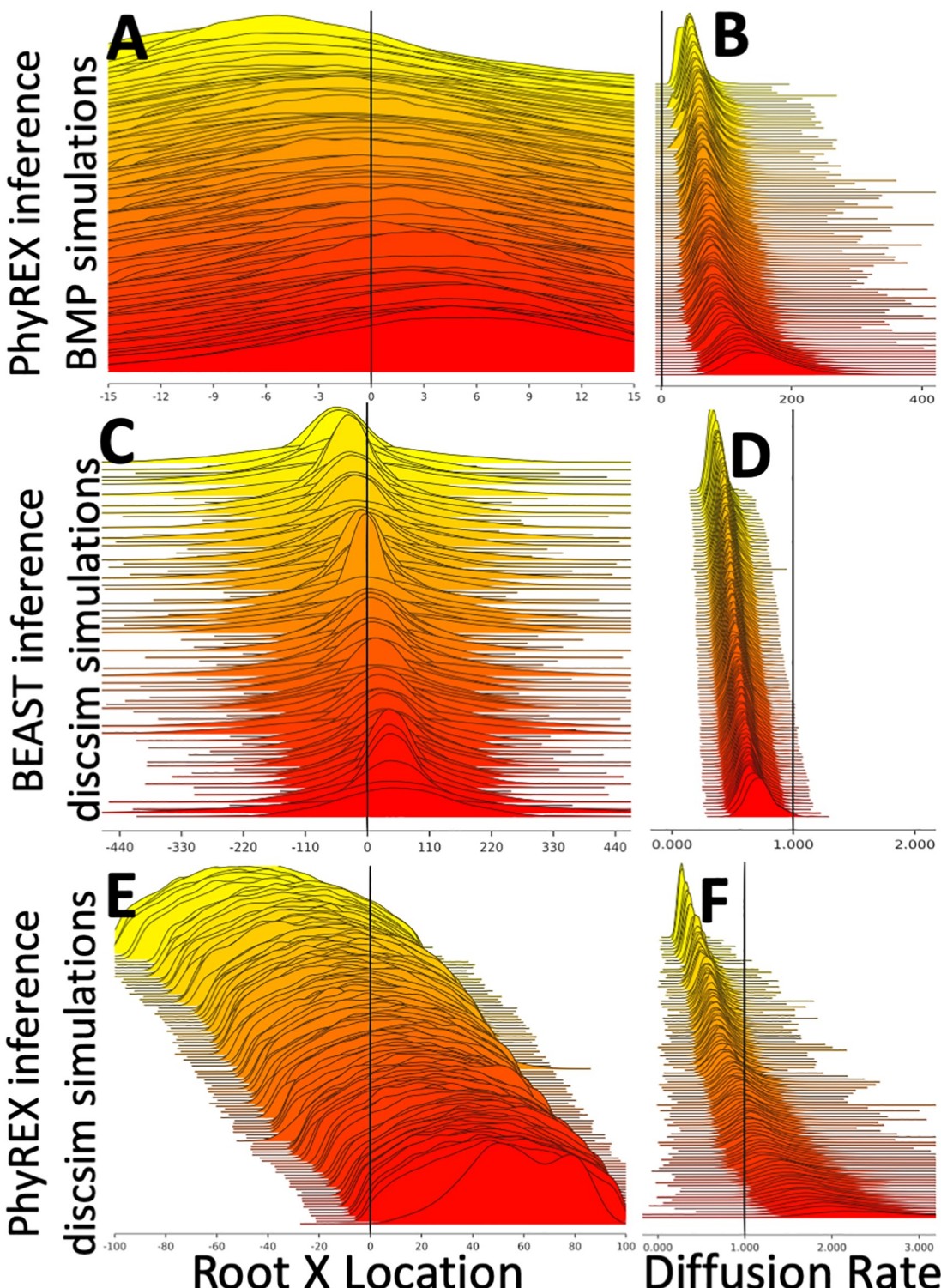

**Fig 3. Comparison of BMP and ΛFV models.** Similarly to Fig 2, here we show posterior distributions of inferred root location and diffusion parameters. In plots **A,B** we show PhyREX inference (which uses the ΛFV model) under BMP simulations with no sampling bias. In plots **C,D** we show BEAST inference (which uses the BMP model) under *discsim* ΛFV simulations with wide sampling. In plots **E,F** we show PhyREX inference under *discsim* wide sampling simulations. Plots **A,C,E** show inference of root X coordinate; plots **B,D,F** show inference of diffusion rate: for BEAST we show the diffusion rate in the X dimension, while for PhyREX we use the diffusion rate calculated using Equation 1 in S1 Text. Here phylogenetic trees were not inferred, but were assumed to be known.

distribution kernels. Another difference is that *discsim* assumes that migration happens on a torus, while PhyREX uses a rectangle (no migration outside the rectangle allowed, representing, for example, the edges of a continent or island). In *discsim*, we always assume a torus of length and width $L = 100$, and in PhyREX we run inference assuming a square space with the same dimensions. We simulated discs of radius $r = 0.1$, impact $u = 0.1$, and event rate $\lambda = \frac{2L^2}{ur^4\pi}$; these parameters were chosen so that migration histories are composed of many small migration events, therefore approximating a Brownian motion, with diffusion rate per dimension approximately $\sigma^2 = 1.0$ (see S1 Text).

We consider two sampling strategies:

- "wide sampling", where 100 samples are collected uniformly at random from the central square of dimensions $50 \times 50$.

- "narrow sampling", where 100 samples are collected uniformly at random from a central square of dimensions $10 \times 10$.

In both sampling strategies, differently from BMP simulations, the diffusion rate was not overestimated by PhyREX (Fig 3F, S14 and S15 Figs). Root location inference in PhyREX is accurate, but posterior intervals usually span most of the simulated geographical range (Fig 3E, S16 and S17 Figs).

When we run BEAST inference on the *discsim* simulations, BMP seems to consistently underestimate the diffusion rate $\sigma^2$ (Fig 3D and S18 Fig). While usually containing the true values, posterior distributions of root locations are even broader than those inferred by PhyREX, and, in particular, broader than the allowed geographical range (Fig 3C and S19 Fig).

These results suggest that the large discrepancies between the simulations under BMP and inference in PhyREX are due to model mis-specification and the inherent differences between the BMP and ΛFV models. In BMP simulations, the very high diffusion rate inferred by PhyREX is likely because the ΛFV model would usually assume that ancestral lineages traverse the considered geographical space several times, backward in time, before finally coalescing, at least in the limit of small and frequent events. The BMP, instead, not assuming endemicity but a rapid spread from an original location, expects shorter distances traveled before lineages find a common ancestor backward in time.

This seems, conversely, also the most plausible reason why the BMP infers low diffusion rate in ΛFV simulations. It seems harder instead to explain why root location posterior distributions inferred by the BMP are broader than those inferred with the ΛFV in ΛFV simulations, while the opposite is true for BMP simulations. A possible reason is that, because the ΛFV assumes a finite space, inferred root locations have to be contained within this space, even if, as typical, lineages are inferred to travel, backward in time, long distances before reaching the root. Under the BMP, in contrast, geographical space is unlimited, and in ΛFV simulations the simulated tree is very long, suggesting long traveled distances from the root to the tips, and therefore high uncertainty in root locations, which more than offsets the effect of sample locations being concentrated inside the ΛFV finite space of interest.

### 3.4 Analysis of a West Nile Virus outbreak

To showcase the importance of these observations with respect to practical epidemiological and phylodynamic investigations, we consider a dataset from a recent West Nile Virus outbreak in North America [14]. We choose this particular dataset due to availability of the data and of clear instructions on how to repeat the published analyses in BEAST https://beast. community/workshop_continuous_diffusion_wnv (accessed on August 2019), reducing the chances of errors on our part. As described in the tutorial, we include sampling time, sampling

location, and genetic sequence data for each sample. We use a separate HKY model for each of the three codon positions, but assume no variation in substitution rates across codons, and we assumed an uncorrelated relaxed molecular clock model [42] with an underlying lognormal distribution. As the tree prior, we employ an exponential growth coalescent model. We assume homogeneous Brownian motion along tree branches.

To investigate the possible effects of sampling bias, we consider two datasets: the first including all samples, and the second including only the western-most half of the samples. This second scenario artificially recreates sampling bias, such as the case where only cases from one half of the country are accessible or considered. This sampling scenario might seem extreme, but it's not uncommon for phylogeographic studies to focus, for example, on a single country or area within a continent, see e.g. [43, 44]. We consider the inference of the location of the root (MRCA) of the western half of the samples. The posterior densities of this same ancestor in the two analyses is very different: when using only western samples, this phylogenetic node is confidently placed in western USA, but when using the whole dataset this same node is confidently placed in the eastern USA instead (Fig 4). Another difference between the two analyses is that when restricting to just the western samples diffusion was inferred to be slower (95% HPD interval [166, 284] km/yr versus [339, 498] in the full analysis).

If we adopt a less extreme degree of sampling bias, for example including 5 or 10 eastern samples in addition to the western ones, we see that, similarly to our simulation results, most of the inference bias is removed (S20 and S21 Figs).

Next, we wanted to see whether, in this scenario, including some sequence-free samples from the eastern side of the country could help in the scenario of biased sampling. To do so, we ran an analysis of the 52 western samples with additionally the 52 eastern samples added as sequence-free samples. These sequence-free eastern samples were included with correct location and sampling time data but without sequence data. In this analysis, the inferred location of the considered node (the MRCA of the western samples) is now shifted eastward, but it is still very different from the inferred location of the same node from the full analysis (Fig 4). It is remarkable that in this dataset, unlike in our simulations, the addition of sequence-free samples does not seem to alleviate the effects of sampling bias very much. One possible explanation for this observation is that, unlike in our BMP simulations, in this case the outbreak seems to migrate westward as time progresses [14], a feature that sequence-free samples are insufficient, in this case, to capture, and that a more specific model might be able to address [45]. This is also hinted at by the fact that performing the same analyses as above but removing the western samples from the full dataset instead of the eastern ones shows almost no effects of the artificially introduced sampling bias (S22 Fig).

Analysing the same datasets with PhyREX also shows different estimates after removing the eastern samples, although this time there is considerable overlap between the different ancestral location estimates and different diffusion rate estimates (Fig 5, S23, S24 and S25 Figs). In principle we would not expect to see considerable differences for different subsampling schemes since the ΛFV model should be robust to sampling biases, as shown in our simulations. This further supports the hypothesis that the progressive westward shift of the outbreak plays a major role in the apparent strong effects of sampling bias in this case. A noticeable difference between BEAST and PhyREX results, also observed in simulations, is that the inferred uncertainty in ancestral location is much larger in PhyREX than in BEAST.

## 3.5 Analysis of a Yellow Fever Virus outbreak

As a second example of real world epidemic analysis, we considered a recent dataset of Yellow Fever Virus (YFV) from Brazil [46]. 65 YFV genomes were collected between January and

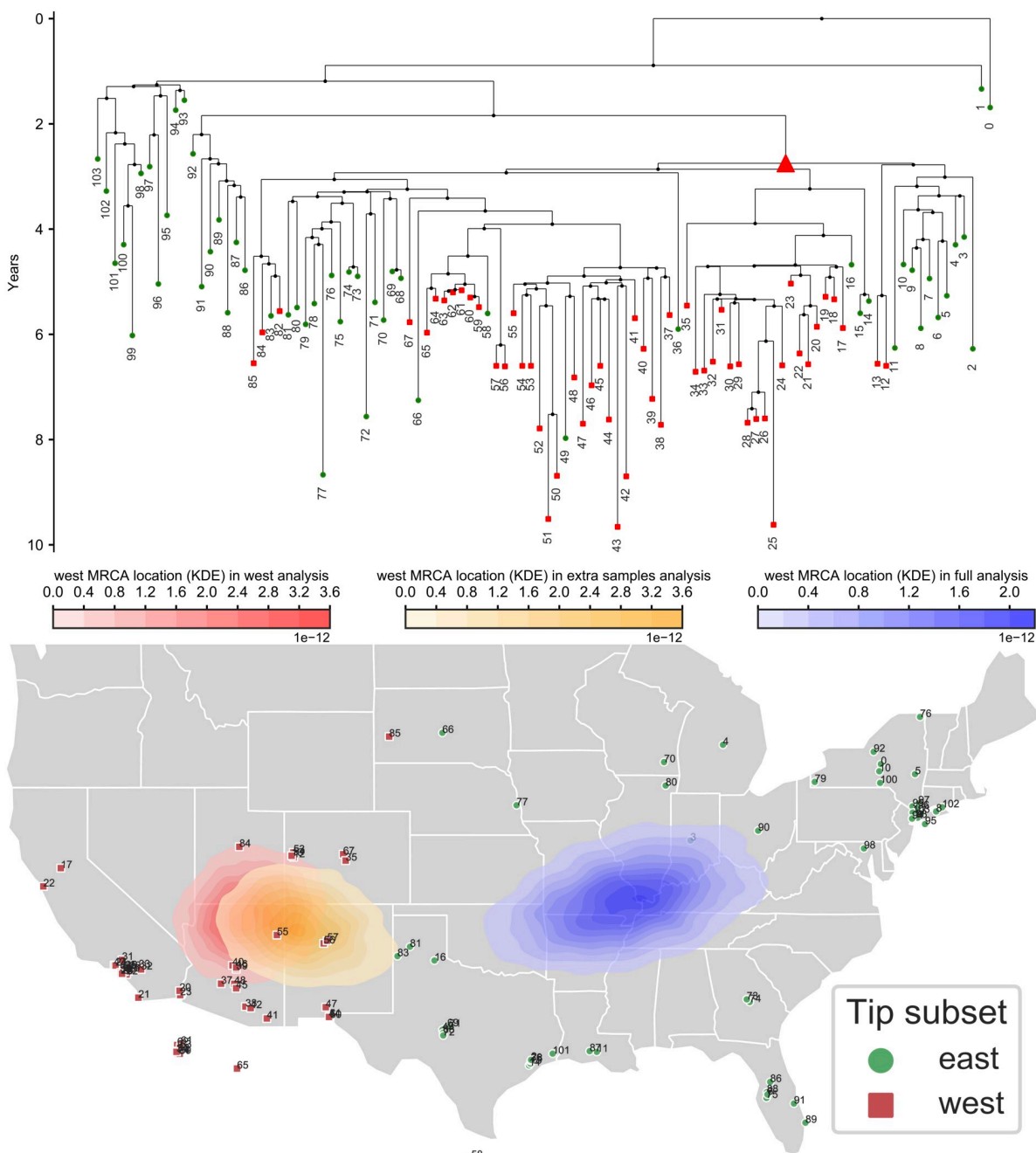

**Fig 4. Recreating the effects of biased sampling over a West Nile Virus outbreak investigation.** We re-analysed the West Nile Virus North America dataset of Pybus and colleagues [14]. At top, we show the maximum clade credibility tree. Branch lengths are in years. Green circles represent eastern samples while red squares represent western samples. The red triangle in the tree represents the node whose location is considered here: the most recent common ancestor (MRCA) of all western samples. Below, the sample locations are shown on a map of the USA. Sample numbers are only used to link samples on the map onto the phylogeny. All three kernel density estimate areas (red, orange and blue) on the map represent the posterior densities of the location of the MRCA of all western samples (red triangle in the phylogeny). The red area represents the posterior from the analysis of only western samples; the blue area is the posterior from the analysis of all samples; the orange area is the posterior from the analysis of the western samples and of sequence-free eastern samples (eastern samples included but without sequence data).

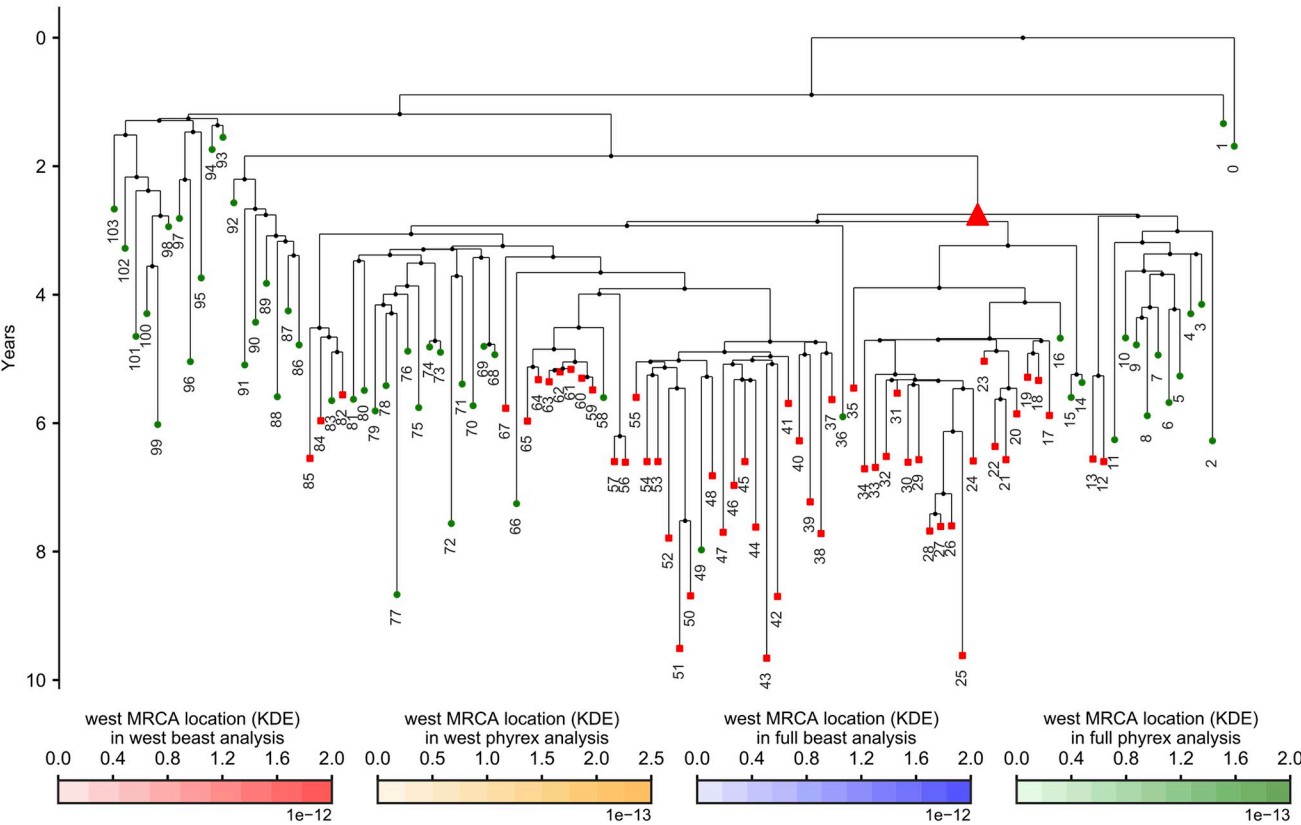

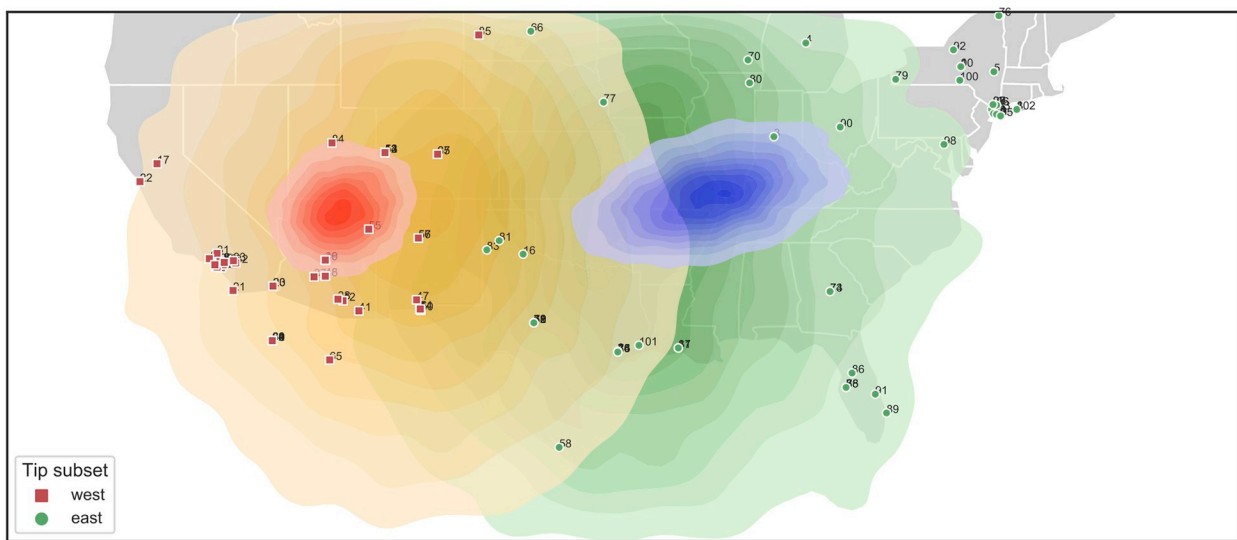

**Fig 5. PhyREX inference and artificial westward sampling bias in a West Nile Virus dataset.** We re-analysed the dataset of Fig 4 using PhyREX for both the full dataset and the one containing only western samples. In PhyREX we defined a rectangular space (outlined in black) with latitude interval [20, 45] and longitude interval [−130, −60]. The red area represents the posterior ancestral location from the analysis of only western samples in BEAST. The orange area is the same for PhyREX. The blue area is the posterior from the analysis of all samples in BEAST. The green area is the posterior from the analysis of all samples in PhyREX. Posterior distribution of the diffusion rate inferred by PhyREX has a mean of 329.9 (95% HPD interval [149.5, 497.4]) km/year in the full dataset and of 133.0 ([15.9, 240.7]) km/year with only western samples.

April 2017, mostly from the Brazilian state of Minas Gerais. Again, we chose this dataset due to availability of data and instructions for repeating the analysis https://beast.community/workshop_continuous_diffusion_yfv (accessed on August 2019). Following the tutorial, we used the same substitution model as for the West Nile Virus dataset, a skygrid coalescent [47] tree prior with 36 grid points, and a Cauchy relaxed random walk model [7].

When recreating sampling bias along a north-south gradient, we find little impact from removing southern samples (S26 Fig), while directional sampling bias seems to have considerable effect in BEAST analyses when removing northern samples (S27 Fig); this bias is greatly reduced by introducing sequence-free samples. PhyREX inference seems, expectedly, mostly unaffected by sampling bias, and shows much broader posterior distributions for ancestral locations (S28 and S29 Figs).

We also observed that many samples of this dataset were collected from few locations: six from Ladainha, five from Novo Cruzeiro, seven from Teófilo Otoni and five from Itambacuri. So, in a second alternative sub-sampling strategy, we reduced the maximum number of samples from any of these locations to two. As before, we aim to artificially recreate different sampling scenarios. We find that, after downsampling, the origin of the outbreak is not inferred anymore to be solely nearby Teófilo Otoni, but also possibly south, close to another cluster of samples near Caratinga (Fig 6 and S30 Fig). A third possible, but low-probability area remains near Belo Horizonte, close to the phylogenetic outgroup location.

These results further suggest that the decision of where to collect samples and which samples to include or exclude from a BMP analysis can significantly impact its conclusions, and that great care should be taken to make sure that the range of samples collected and their density reflect real geographic distributions.

## 4 Conclusion

We have shown that continuous space phylogeographic inference can be negatively affected by sampling biases, such as sampling efforts being focused in certain areas over others. These biases can lead to strongly mis-inferred ancestral node locations, up to completely excluding the true origin of outbreaks with complete confidence. These biases also usually lead to underestimating the dispersal velocity of pathogens, and can in some cases lead to inference of artificial patterns of correlated spread across space dimensions.

We explored possible ways to tackle these issues. A possible approach is to include sequence-free samples, which correspond to known cases (for which we know date and location) which have no corresponding genetic information. We find that sequence-free samples can considerably improve inference and compensate sampling biases, but that in most scenarios it would be computationally unfeasible or unrealistic to completely eliminate the effects of sampling biases, if possible at all. We confirm these results on real datasets from West Nile Virus and Yellow Fever Virus outbreaks by artificially recreating scenarios of sampling bias.

As an alternative, we investigated the use of an inference model that is in theory not affected by sampling biases: the ΛFV implemented in PhyREX. Similarly to the structured coalescent, in fact, the ΛFV model conditions on sampling locations, and so should not be adversely affected when samples are not collected proportionally to prevalence; note however that the ΛFV might be affected by geographical biases in the case where the true geographical range of the organism being studied is not known, but we don't focus on this aspect here. We confirm that indeed this model is seemingly unaffected by different sampling strategies, but, more importantly, the model is also very different from the BMP, resulting in very different estimates. The assumptions and applicability of these two models being so different, we would expect few scenarios of common applicability. The BMP, in fact, well-describes the spread of

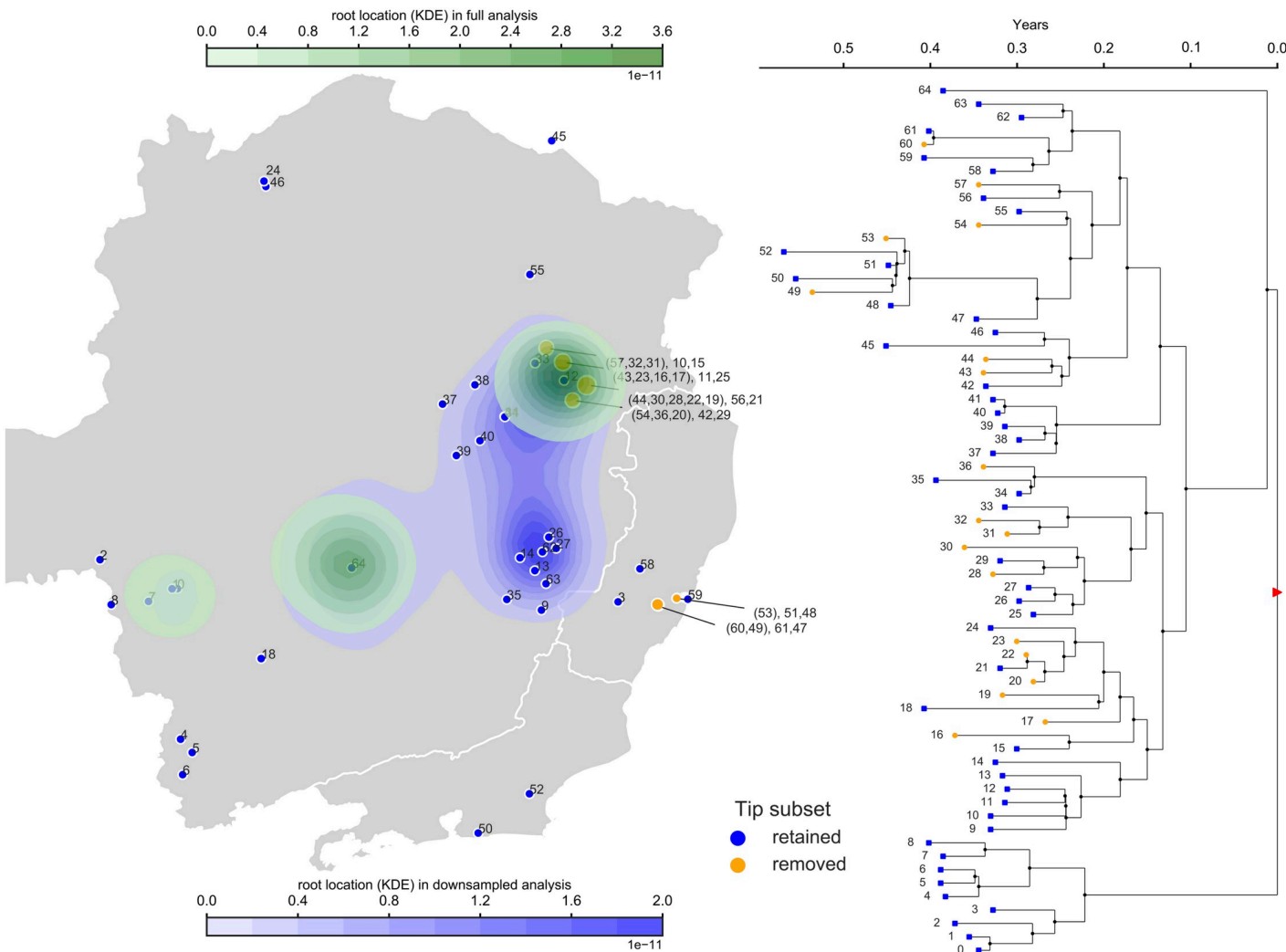

**Fig 6. Phylogeographic analyses of Yellow Fever Virus in Brazil and effects of location over/under-sampling.** Here we compare the results of BMP analysis using all the data from [46], versus reducing the number of samples from each location to allow maximum two samples per location. The map on the left shows the Brazilian states of Minas Gerais (center), Rio de Janeiro (south-east) and Espírito Santo (east); blue samples are the ones used in both analyses, while orange circles represent locations that were downsampled—numbers in parentheses are the samples that were removed in the downsampled analysis. The blue area on the map shows the inferred posterior distribution (kernel density estimate) of root location in the full analysis; the green area shows the posterior distribution of root location in the downsampled analysis. On the right is the maximum clade credibility phylogeny inferred from the analysis of the whole dataset. Orange tips are the ones that were removed in the downsampled dataset; the red triangle marks the root node.

outbreak within a new, unlimited environment, or at least within an area that is large compared to the current range of the outbreak. For example, in BMP simulations, lineages generally spread out from the original source and move in all directions, on average spreading further away from the origin as time progresses. The ΛFV, instead, fits better a scenario where an outbreak (or any population) has become endemic within an area, or at least where lineages are expected to have migrated across the area since their introduction. For example, in ΛFV simulations lineages usually tend to cross the considered geographic space several times before they all find a common ancestor. One practical consequence of this is that inference under the ΛFV of possible root locations can be, on the same data, much broader than the same inference under the BMP. Similarly, diffusion rates estimated under the ΛFV can be much higher than those estimated under the BMP. It is possible, however, that, in some scenarios or with

some modifications, these two models would more substantially overlap in applicability. An example could be when restricting the allowed geographic range within the BMP to a limited space, that is, not allowing BMP migration outside of a confined area. In fact, we suspect that simulating migration under such a version of the BMP, and simulating a long phylogeny (in terms of distance traveled from the root before samples are collected) would lead to patterns very close to those simulated under the ΛFV. Another possibility is that considering samples collected at different points in time would reduce uncertainty in the inference under both models.

In the future, it would be of great interest to make the BMP robust to sampling biases by conditioning the BMP geographical likelihood on the location of collected samples, or to consider alternative, possibly approximate models that share this property, which is a topic we are currently working on. While in this manuscript we have only considered a simple model of migration, that is with no directional bias and no variation in diffusion rate over time, location or lineages, it will be interesting in the future to investigate how the relaxation of these assumptions [7, 45] would impact the results presented here. Another issue that we think would be important to address in the future is the elevated computational demand and slow convergence of inference under the ΛFV model; further work on this model, for example in the form of efficient approximations, could greatly improve its applicability in practical scenarios. While the simulation scenarios considered here are quite extreme in terms of sampling biases, it will be important to consider the degree and type of sampling biases in real life outbreaks (see e.g. [48]) in future research.

In conclusion, we report that often the choice of model and of sampling strategy has dramatic effects on the results of a continuous phylogeographic analysis. We therefore recommend attention be paid when deciding a sampling strategy for BMP so that the range and distribution of collected samples would reflect the geographical distribution of the outbreak as much as possible. We also recommend an appropriate phylogeographic model to be used, depending on the history and range of the considered outbreak.

## Supporting information

**S1 Text. Supplementary text.** The Supplement contains additional information regarding the methods used.
(PDF)

**S1 Fig. Effects of sampling bias on BMP root location inference.** Here a BMP model was used both for simulation and inference. Plots show inferred posterior distributions for the X dimension position of the tree root (plots **A,C,E,G**), and its Y dimension position (plots **B,D, F,H**). In each plot, the 100 distributions represent 100 independent replicates, and are vertically sorted based on the posterior median. Vertical black lines show the true, simulated values (in this case always 0). Plots **A,B** are from simulations with non-biased samples, plots **C,D** with "Central" biased samples, plots **E,F** with "Diagonal" biased samples, and plots **G,H** with "One-sided" sampling bias. Since in many cases the MRCA of the collected samples is not the root of the whole simulated phylogeny (which was simulated at location (0, 0)), in each simulation all locations are translated (in mathematical sense) so that the true simulated sample MRCA is always at (0, 0).
(PNG)

**S2 Fig. Effects of sampling bias on BMP inference of diffusion parameters.** Here a BMP model was used both for simulation and for inference. Plots show inferred posterior distributions for the diffusion rate in the X dimension (plots **A,D,G,J**), in the Y dimension (plots **B,E,**

**H,K**), and for the correlation between the diffusion in the two dimensions (plots **C,F,I,L**). In each plot, the 100 distributions represent 100 independent replicates, and are vertically sorted based on the posterior median. Vertical black lines show the true, simulated values (in this case 1 for rates and 0 for the correlation). Plots **A-C** are from simulations with non-biased samples, plots **D-F** with "Central" biased samples, plots **G-I** with "Diagonal" biased samples, and plots **J-L** with "One-sided" sampling bias.
(PNG)

**S3 Fig. Effects of sampling bias on BMP inference with no phylogenetic uncertainty.** Here a BMP model was used both for simulation and inference, and the phylogenetic tree is assumed to be known without uncertainty. Plots show inferred posterior distributions for the X dimension position of the tree root (plots **A,D**), the diffusion rate along the X dimension (plots **B,E**), and the correlation between the diffusion in the two dimensions (plots **C,F**). In each plot, the 100 distributions represent 100 independent replicates, and are vertically sorted based on the posterior median. Plots **A-C** are from simulations with non-biased samples. Plots **D,E,F** are respectively with "One-sided" sampling bias, "Central" sampling bias, and "Diagonal" sampling bias.
(PNG)

**S4 Fig. Effects of sampling bias on BMP root location inference, no phylogenetic uncertainty.** Similarly to S1 Fig, here we show BMP inference of root locations under BMP simulations, but this time the phylogenetic tree is assumed to be known without uncertainty.
(JPG)

**S5 Fig. Effects of sampling bias on BMP inference of diffusion parameters, no phylogenetic uncertainty.** Similarly to S2 Fig, here we show BMP inference of diffusion parameters under BMP simulations, but this time the phylogenetic tree is assumed to be known without uncertainty.
(JPG)

**S6 Fig. Effects of varying intensities of sampling bias on BMP inference.** A BMP model was used both for simulation of 10,000 cases and inference from 100 samples. Plots show inferred posterior distributions for the X dimension position of the tree root (plots **A,C,E,G,I**) and the diffusion rate along the X dimension (plots **B,D,F,H,J**). In each plot, the 100 distributions represent 100 independent replicates, and are vertically sorted based on the posterior median. Plots **A,B** are from simulations with non-biased samples, while plots **I,J** are from simulations where all samples are biased. The other plots show intermediate levels of sampling bias, where 25% (plots **C,D**), 50% (plots **E,F**), and 75% (plots **G,H**) of samples are respectively collected at the positive extreme end of the X axis range, while the remaining samples are randomly selected.
(JPG)

**S7 Fig. Effects of extra sequence-free samples on BMP root location inference.** Similarly to S1 Fig, here we show BMP inference of root locations under BMP simulations, but this time we include 50 extra sequence-free samples (without genetic sequence but with correct date and sampling location).
(JPG)

**S8 Fig. Effects of extra samples on BMP inference of diffusion parameters.** Similarly to S2 Fig, here we show BMP inference of diffusion parameters under BMP simulations, but this time we include 50 extra sequence-free samples (without genetic sequence but with correct

date and sampling location).
(JPG)

**S9 Fig. Effects of number of extra sequence-free samples on BMP inference.** Here we simulate under a BMP model and "One-sided" sampling bias, but only simulate 100 cases and sample 20 of them. **A-B** Show inference of root X location and diffusion rate in the X dimension respectively. **C, E, G, I** show inference of root X location after adding some (respectively 20, 40, 60, and 80) of the 80 non-sampled cases to the analysis as sequence-free samples. **D, F, H, J** show inference of diffusion rate in the X dimension in the same scenarios.
(JPG)

**S10 Fig. Root location inference with the ΛFV under BMP simulations.** Similarly to S1 Fig, here we show inference of root locations under BMP simulations, but this time inference is performed under the ΛFV model implemented in PhyREX.
(JPG)

**S11 Fig. Root location inference with the ΛFV under BMP simulations, only converged runs.** Similar to S10 Fig, but showing only converged MCMC runs (where all considered parameters have ESS>100).
(JPG)

**S12 Fig. Inferred diffusion rate with the ΛFV under BMP simulations.** Similarly to S2 Fig, here we show inference of diffusion parameters under BMP simulations, but this time inference is performed under the ΛFV model implemented in PhyREX. Plots **A,E,I,M** show inferred diffusion rate $\sigma^2$ using Equation 1 in S1 Text, plots **B,F,J,N** use method "dispersion across short distance from the tips", plots **C,G,K,O** use method "dispersion across long distance from the tips", and plots **D,H,L,P** use method "dispersion from the root"; see S1 Text for more details.
(JPG)

**S13 Fig. Inferred diffusion rate with the ΛFV under BMP simulations, only converged runs.** Similar to S12 Fig, but showing only converged MCMC runs (where all considered parameters have ESS>100).
(JPG)

**S14 Fig. Inferred diffusion rate with the ΛFV under ΛFV simulations.** Similarly to S12 Fig, here we show PhyREX inference of diffusion parameters, but this time simulations are performed under the ΛFV model implemented in *discsim*.
(PNG)

**S15 Fig. Inferred diffusion rate with the ΛFV under ΛFV simulations, only converged runs.** Similar to S14 Fig, but showing only converged MCMC runs (where all considered parameters have ESS>100).
(JPG)

**S16 Fig. Root location inference with the ΛFV under ΛFV simulations.** Similarly to S10 Fig, here we show PhyREX inference of root locations, but this time simulations are performed under the ΛFV model implemented in *discsim*.
(PNG)

**S17 Fig. Root location inference with the ΛFV under ΛFV simulations, only converged runs.** Similar to S16 Fig, but showing only converged MCMC runs (where all considered parameters have ESS>100).
(PNG)

**S18 Fig. Inference of diffusion parameters using the BMP from simulations under the ΛFV model.** Here the BEAST BMP model was used for inference, while the *discsim* ΛFV model was used for inference. Plots show inferred posterior distributions for the diffusion rate in the X dimension (plots **A,D**), in the Y dimension (plots **B,E**), and for the correlation between the diffusion in the two dimensions (plots **C,F**). In each plot, the 100 distributions represent 100 independent replicates, and are vertically sorted based on the posterior median. Vertical black lines show the true, simulated values (in this case 1 for rates and 0 for the correlation). Plots **A-C** are from simulations with wide sampling, while plots **D-F** are with narrow sampling.
(PNG)

**S19 Fig. Inference of root location using the BMP from simulations under the ΛFV model.** Here the BEAST BMP model was used for inference, while the *discsim* ΛFV model was used for inference. Plots show inferred posterior distributions for the X dimension position of the tree root (plots **A,C**), and its Y dimension position (plots **B,D**). In each plot, the 100 distributions represent 100 independent replicates, and are vertically sorted based on the posterior median. Vertical black lines show the true, simulated values (in this case always 0). Plots **A,B** are from simulations with wide sampling, while plots **C,D** are from simulations with narrow sampling. Since in many cases the MRCA of the collected samples is not the root of the whole simulated phylogeny (which was simulated at location (0, 0)), in each simulation all locations are translated (in mathematical sense) so that the true simulated sample MRCA is always at (0, 0).
(PNG)

**S20 Fig. Recreating the effects of strong biased sampling over a West Nile Virus outbreak investigation.** Similarly as in Fig 4, we recreate sampling bias in a West Nile Virus dataset by comparing the inference between the full dataset and the same dataset after excluding eastern samples. For all analyses we compare the inference of the ancestral location of the MRCA of all western samples. Here, we consider three analyses: the one with all samples (blue kernel density on the map), the one with only western samples (red kernel density), and one with western samples plus 5 random eastern samples (pink kernel density) representing a scenario of strong, but not extreme, sampling bias. The 5 random eastern samples are also represented in the phylogeny as pink tips.
(PDF)

**S21 Fig. Recreating the effects of moderate biased sampling over a West Nile Virus outbreak investigation.** Same as S20 Fig, but this time with 10 extra eastern samples instead of 10 (yellow tips and orange kernel density), representing a scenario of moderate sampling bias.
(PDF)

**S22 Fig. Recreating eastward biased sampling in a West Nile Virus outbreak investigation.** We re-analysed the West Nile Virus North America dataset of Pybus and colleagues [14] as in the main text, but this time selecting only the east-most samples. At top, we show the maximum clade credibility tree from the full dataset. Branch lengths are in years. Green circles represent western samples while red squares represent eastern ones. The red triangle in the tree represents the node whose location is considered here: the most recent common ancestor (MRCA) of all samples. Below, the sample locations are shown on a map of the USA. Sample numbers are only used to link samples on the map to samples on the phylogeny. All three kernel density estimate areas (red, orange and blue) on the map represent the posterior densities of the location of the MRCA (red triangle in the phylogeny). The red area represents the posterior from the analysis of only eastern samples; the blue area is the posterior from the analysis

of all samples; the orange area is the posterior from the analysis of the eastern samples and of sequence-free western samples (western samples included but without sequence data).
(PDF)

**S23 Fig. PhyREX inference and artificial westward sampling bias in a West Nile Virus dataset, with a broader space.** Same analysis as in Fig 5, but using a broader rectangular space (outlined in black) in PhyREX, latitude interval [6, 50] and longitude interval [−140, −35]. Posterior distribution of the diffusion rate inferred by PhyREX has a mean of 389.2 (95% HPD interval [132.4, 642.3]) km/year in the full dataset and of 150.5 ([16.6, 287.8]) km/year with only western samples.
(PDF)

**S24 Fig. PhyREX inference and artificial eastward sampling bias in a West Nile Virus dataset.** We re-analysed the dataset of S22 Fig, using PhyREX for both the full dataset and the one containing only eastern samples. In PhyREX we defined a rectangular space (outlined in black) with latitude interval [20, 45] and longitude interval [−130, −60]. The red area represents the posterior ancestral location from the analysis of only eastern samples in BEAST. the orange area is the same for PhyREX. The blue area is the posterior from the analysis of all samples in BEAST. The green area is the posterior from the analysis of all samples in PhyREX. Posterior distribution of the diffusion rate inferred by PhyREX has a mean of 329.9 (95% HPD interval [149.5, 497.4]) km/year in the full dataset and of 219.4 ([39.3, 391.5]) km/year with only eastern samples.
(PDF)

**S25 Fig. PhyREX inference and artificial eastward sampling bias in a West Nile Virus dataset, with a broader space.** Same analysis as in S24 Fig, but using a broader rectangular space (outlined in black) in PhyREX, latitude interval [6, 50] and longitude interval [−140, −35]. Posterior distribution of the diffusion rate inferred by PhyREX has a mean of 389.2 (95% HPD interval [132.4, 642.3]) km/year in the full dataset and of 304.7 ([36.2, 580.4]) km/year with only eastern samples.
(PDF)

**S26 Fig. Recreating the effects of biased sampling over a Yellow Fever Virus outbreak investigation.** To investigate the possible effects of sampling bias, we again consider the effects or restricting an analysis to a geographical subsample of an original dataset. Here we compare the results of BMP analysis using all the data from [46] versus using only the northern samples (latitude above −19.0, red squares in the phylogeny and on the map). On top is the maximum clade credibility phylogeny inferred from analysing the whole dataset. On the map (bottom) we show the location of the samples and the inferred location of the most recent common ancestor of all southern samples (red triangle in the phylogeny). The three, almost completely overlapping colored areas on the map show the inferred posterior distribution (kernel density estimate) of the location of this ancestor from three analyses: using only the northern samples (red area), using all samples (blue area) or using only the northern samples but adding the southern ones as sequence-free samples (orange areas). The three small areas completely overlap, masking each other in the figure. A noticeable difference between the analyses is that when restricting to just the northern samples diffusion was inferred to be slower (95% HPD interval [152, 1018] km/yr versus [471, 1512] of the full analysis).
(PDF)

**S27 Fig. Southward biased sampling in a Yellow Fever Virus outbreak investigation.** Complementarily to S26 Fig, we consider the effects or restricting an analysis to a southern

geographical subsample of the YFV dataset. We compare the results of BMP analysis using all the data from [46] versus using only the southern samples (latitude below −19.0, red squares in the phylogeny and on the map). On top is the maximum clade credibility phylogeny inferred from analysing the whole dataset. On the map (bottom) we show the location of the samples and the inferred location of the most recent common ancestor (red triangle in the phylogeny). The three colored areas on the map show the inferred posterior distribution (kernel density estimate) of the root location from three analyses: using only the southern samples (red area), using all samples (blue area) or using only the southern samples but adding the northern ones as sequence-free samples (orange areas). Due to difficulties in convergence, and following results from the analysis with all samples, in the analysis with sequence-free samples we added a normal distribution prior over root height with mean 0.7 and standard deviation 0.25.
(PDF)

**S28 Fig. PhyREX inference and Northward biased sampling in a Yellow Fever Virus dataset.** Similarly to S26 Fig, we consider the effects or restricting an analysis to a northern geographical subsample of the YFV dataset, and we compare PhyREX and BEAST inference. The colors on the map show the posterior distribution of the location of the considered ancestor for the analysis with BEAST and northern samples (red), PhyREX and northern samples (orange), BEAST and all samples (blue), and PhyREX and all samples (green). In PhyREX we defined a rectangular space with latitude interval [−23, −15] and longitude interval [−48, −40]. Posterior distribution of the diffusion rate inferred by PhyREX has a mean of 537.0 (95% HPD interval [56.8, 1030.3]) km/year in the full dataset and of 1132.2 ([68.4, 2373.6]) km/year with only northern samples.
(PDF)

**S29 Fig. PhyREX inference and Southward biased sampling in a Yellow Fever Virus dataset.** Similarly to S27 Fig, we consider the effects or restricting an analysis to a southern geographical subsample of the YFV dataset, and we compare PhyREX and BEAST inference. The colors on the map show the posterior distribution of the location of the considered ancestor for the analysis with BEAST and southern samples (red), PhyREX and southern samples (orange), BEAST and all samples (blue), and PhyREX and all samples (green). In PhyREX we defined a rectangular space with latitude interval [−23, −15] and longitude interval [−48, −40]. Posterior distribution of the diffusion rate inferred by PhyREX has a mean of 537.0 (95% HPD interval [56.8, 1030.3]) km/year in the full dataset and of 825.6 ([66.6, 1562.5]) km/year with only southern samples.
(PDF)

**S30 Fig. Effects of location over/under-sampling and use of PhyREX on YFV dataset.** Here we compare the results of using all the YFV dataset versus downsampling each location up to allowing a maximum of two samples per location, similarly to Fig 6. Here we also compare BEAST and PhyREX inference. The green area on the map shows the inferred posterior distribution (kernel density estimate) of root location in the full analysis in BEAST, while red is the same for PhyREX. The blue area shows the posterior distribution of root location in the downsampled analysis in BEAST, while cyan is the same for PhyREX. In PhyREX we defined a rectangular space with latitude interval [−23, −15] and longitude interval [−48, −40]. Posterior distribution of the diffusion rate inferred by PhyREX in the sownsampled analysis has a mean of 537.1 (95% HPD interval [60.8, 1049.4]) km/year.
(PDF)

## Acknowledgments

We thank the developers of the script newick.py (https://github.com/tyjo/newick.py), of which we used an adaptation in our simulations.

## Author Contributions

**Conceptualization:** Nicola De Maio.

**Data curation:** Yuxuan Sun, Guy Baele, Stephane Guindon, Nicola De Maio.

**Formal analysis:** Antanas Kalkauskas, Yuxuan Sun, Stephane Guindon, Nicola De Maio.

**Funding acquisition:** Antanas Kalkauskas, Yuxuan Sun, Nick Goldman, Nicola De Maio.

**Investigation:** Antanas Kalkauskas, Yuxuan Sun, Guy Baele, Stephane Guindon, Nicola De Maio.

**Methodology:** Antanas Kalkauskas, Guy Baele, Stephane Guindon, Nicola De Maio.

**Project administration:** Nick Goldman, Nicola De Maio.

**Software:** Antanas Kalkauskas, Umberto Perron, Stephane Guindon, Nicola De Maio.

**Supervision:** Nick Goldman, Nicola De Maio.

**Validation:** Antanas Kalkauskas, Nicola De Maio.

**Visualization:** Antanas Kalkauskas, Umberto Perron, Nicola De Maio.

**Writing – original draft:** Nicola De Maio.

**Writing – review & editing:** Antanas Kalkauskas, Umberto Perron, Yuxuan Sun, Nick Goldman, Guy Baele, Stephane Guindon, Nicola De Maio.

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
