## [Decision Letter · Decision Letter 0]

3 Jul 2020

Dear Dr De Maio,

Thank you very much for submitting your manuscript "Sampling bias and model choice in continuous phylogeography: getting lost on a random walk." for consideration at PLOS Computational Biology. As with all papers reviewed by the journal, your manuscript was reviewed by members of the editorial board and by several independent reviewers. The reviewers appreciated the attention to an important topic. Based on the reviews, we are likely to accept this manuscript for publication, providing that you modify the manuscript according to the review recommendations.

Sincerely,

Sergei L. Kosakovsky Pond, PhD

Associate Editor

PLOS Computational Biology

Stefano Allesina

Deputy Editor

PLOS Computational Biology

[LINK]

Reviewer's Responses to Questions

**Comments to the Authors:**

Reviewer #1: In their paper, the authors present a timely and important study into the effects of sampling bias on two popular continuous phylogeography models. I found the topic compelling, and the paper to be generally well written. As such I have no major objection to its being published immediately as it stands.

That said, I do have several small comments/questions for the authors that they may wish to address/clarify in order to improve the submission.

1. In lines 316-318 on page 9, the authors state that the inclusion of unsequenced samples doesn't completely eliminate the bias due to nonrandom sampling, saying that to achieve this one would "probably" have to include enough additional samples to make the total distribution of samples even. How difficult would it be to show that this does _actually_ occur, even for a small example?

2. I'm slightly worried about the convergence criterion for the PhyREX analyses detailed on lines 334-336 on page 10. These lines do not exclude the possibility that analyses with arbitrarily low ESS values have been included in the analysis. Maybe a threshold ESS could be instigated, below which analyses are excluded? Although of course one needs to be careful there too, as such thresholds can potentially bias results too. Otherwise, would it be possible simply to run the low-ESS analyses for another week or two?

I'm not _really_ concerned that the inference results are biased due to lack of convergence, but being able definitively say that all analyses were run to some minimum ESS would remove any shred of doubt.

3. On lines 354-356 on page 10, the authors say that using a prior to limit the size of the ΛFV radius parameter improves the agreement between the inferred diffusion rate under this model and the true value, as simulated under the BMP. It's intuitively clear why the bias is present and why this trick helps, but I wonder whether another approach might be to define some kind of (approximate) effective diffusion rate on the ΛFV side which would be more directly comparable to the diffusion rate on the BMP side. The reason I'm suggesting this is that the existing "bias" seems more just a result of failing to compare apples with apples.

4. On lines 358-359 on page 10, the authors describe their ΛFV simulation study setup. It's not clear to me why they chose to simulate trees under the slightly different ΛFV implementation available in discsim, as opposed to precisely the model that the inference is done under. This seems like a needless complication of the study method.

Typographical errors:

1. Line 18, "include parameter" -> "include parameters"

2. Line 86, "top node" could be ambiguous

3. Line 104, "nominator" -> "numerator"

4. Lines 466-468 appear to be a broken sentence.

Reviewer #2: This a timely and mostly straightforward paper. I found a few issues that need clarification, and think more realistic sampling bias scenarios would greatly improve its impact, but as is I think it’s a useful and somewhat disconcerting paper for the utility of phylogeographic models in estimating epidemiological parameters in outbreaks. If anything I think the authors are too positive about the performance of these methods given nonrandom sampling.

Essentially, the authors summarize two continuous phylogeography models and compare their robustness to biased sampling when attempting to infer the origin of an outbreak and its dispersal rate. Given the current pandemic this is an important area of research. I found the headline findings not particularly surprising – BMP is badly affected by biased sampling, spatial-lambda-fleming-viot (SLFV for the remainder of this review) less so – and the suggestion to use sequence-free samples in BMP is well described and useful. However I was surprised how poorly the SLFV model performed, think the bias scenarios tested are too extreme, and would like to see a little more discussion of what sampling bias looks like in real data. Here I’ll give some broader suggestions for improving the paper, but I don’t think they’re necessarily required for publication – just that they could improve it (or possibly could help guide future work).

Comments:

1. I think the authors overlook a significant issue when claiming the SLFV is not affected by sampling bias: the definition of the landscape boundaries required for inference under the SLFV will itself be shaped by the spatial distribution of samples in realistic settings.

2. This is connected to a second issue – the biased sampling scenarios tested are more extreme than I expect in a real data-deficient outbreak situation (as we are currently facing). Tests in which sampling is uneven relative to population density over space rather than completely excluding much of the map would be more relevant to real-world data analysis. What we see with covid or flu is an excess of samples in rich countries and a deficit in poor countries, relative to local viral populations, and also a severe local clustering within countries as a few hospitals do most of the sequencing. I suspect this has quite different effects on these analyses than the more extreme forms of sampling bias tested here. These scenarios are certainly useful as hyperbolic cases, but more realistic scenarios would be more impactful. However I think that is beyond the scope of a revision to the current manuscript (though I'd love to see the authors pursue it in the future!), so at least briefly discussing the sorts of bias in real-world data would be useful.

3. (Line 42 – 53) It seems important here to note that the structured coalescent is a discrete population model so requires binning populations, which forces additional ad-hoc sample choices by users. In addition, though the structured coalescent is robust to uneven sampling across populations, I don’t think it is at all robust to missing samples from one “population” entirely (which seems more similar to the sampling designs tested here). Note this issue of entirely missing a population also impacts SLFV through the choice of the landscape boundaries.

4. In general the results of the SLFV model are, to me, surprisingly bad – given these parameter settings and empirical data, the uncertainty on root location is so broad as to provide essentially no useful information. Do the authors agree with this assessment? Is there a setting in which PhyREX does perform well? I’d like a little more clarity on this case. Perhaps the authors could discuss PhyREX’s own validation data where it presumably performs reasonably well?

5. At least one figure showing PhyREX performance on empirical data should be included in the main text, since that comparison is a major focus of the paper.

Reviewer #3: In this paper, the authors study the effect of sampling bias on two phylogeographic models: Brownian Motion and Lambda-Fleming-Viot. The simulations and two real data analyses are well-designed for illustrating the impact of sampling bias. However, I find that the scenarios considered in the paper are too extreme. For example, in the analysis of a West Nile Virus Outbreak, the sampling bias dataset include only the western-most half of the samples. A less extreme scenario such that 80% of the data points are in the west while the remaining data points spread out may be closer to a real dataset. Therefore, I suggest the authors adding these scenarios to their studies (in both simulations and data analyses).

**Have all data underlying the figures and results presented in the manuscript been provided?**

Reviewer #1: Yes

Reviewer #2: Yes

Reviewer #3: Yes

PLOS authors have the option to publish the peer review history of their article (what does this mean?). If published, this will include your full peer review and any attached files.

Reviewer #1: No

Reviewer #2: No

Reviewer #3: No
---

## [Editor Report · Decision Letter 1]

24 Nov 2020

Dear Dr De Maio,

We are pleased to inform you that your manuscript 'Sampling bias and model choice in continuous phylogeography: getting lost on a random walk.' has been provisionally accepted for publication in PLOS Computational Biology.

Best regards,

Sergei L. Kosakovsky Pond, PhD

Associate Editor

PLOS Computational Biology

Stefano Allesina

Deputy Editor

PLOS Computational Biology

---

## [Editor Report · Acceptance letter]

29 Dec 2020

PCOMPBIOL-D-20-00292R1 

Sampling bias and model choice in continuous phylogeography: getting lost on a random walk.

Dear Dr De Maio,

I am pleased to inform you that your manuscript has been formally accepted for publication in PLOS Computational Biology. Your manuscript is now with our production department and you will be notified of the publication date in due course.

With kind regards,

Livia Horvath
